# Assessing the Safety and Therapeutic Efficacy of Cannabidiol Lipid Nanoparticles in Alleviating Metabolic and Memory Impairments and Hippocampal Histopathological Changes in Diabetic Parkinson’s Rats

**DOI:** 10.3390/pharmaceutics16040514

**Published:** 2024-04-08

**Authors:** Sarawut Lapmanee, Sakkarin Bhubhanil, Prapimpun Wongchitrat, Natthawut Charoenphon, Anjaree Inchan, Thitaphat Ngernsutivorakul, Piroonrat Dechbumroong, Mattaka Khongkow, Katawut Namdee

**Affiliations:** 1Department of Basic Medical Sciences, Faculty of Medicine, Siam University, Bangkok 10160, Thailand; sarawut.lap@siam.edu (S.L.); sakkarin.bhu@siam.edu (S.B.); 2Center for Research Innovation and Biomedical Informatics, Faculty of Medical Technology, Mahidol University, Nakon Pathom 73170, Thailand; prapimpun.won@mahidol.ac.th; 3Department of Anatomy, Faculty of Medical Science, Naresuan University, Phitsanulok 65000, Thailand; natthawutch@nu.ac.th; 4Department of Physiology, Faculty of Medical Science, Naresuan University, Phitsanulok 65000, Thailand; anjaree.in@gmail.com; 5Department of Chemistry, Faculty of Science, Kasetsart University, Bangkok 10903, Thailand; thitaphat.n@ku.ac.th; 6National Nanotechnology Centre (NANOTEC), National Science and Technology Development Agency, Pathumthani 12120, Thailand; piroonrat.d@gmail.com (P.D.); mattaka@nanotec.or.th (M.K.)

**Keywords:** cannabidiol, drug delivery, lipid nanoparticles, Parkinson’s disease, type 2 DM

## Abstract

Diabetic Parkinson’s disease (DP) is a progressive neurodegenerative disease with metabolic syndrome that is increasing worldwide. Emerging research suggests that cannabidiol (CBD) is a neuropharmacological compound that acts against this disease, especially CBD in nano-formulation. The safety of cannabidiol lipid nanoparticles (CBD-LNP) was evaluated by assessing in vitro cytotoxicity in neurons and therapeutic outcomes in a DP animal model, including metabolic parameters and histopathology. CBD-LNPs were fabricated by using a microfluidization technique and showed significantly lower cytotoxicity than the natural form of CBD. The DP rats were induced by streptozotocin followed by a 4-week injection of MPTP with a high-fat diet. Rats were treated orally with a vehicle, CBD, CBD-LNP, or levodopa for 4 weeks daily. As a result, vehicle-treated rats exhibited metabolic abnormalities, decreased striatal dopamine levels, and motor and memory deficits. CBD-LNP demonstrated reduced lipid profiles, enhanced insulin secretion, and restored dopamine levels compared to CBD in the natural form. CBD-LNP also had comparable efficacy to levodopa in ameliorating motor deficits and memory impairment in behavior tests. Interestingly, CBD-LNP presented migration of damaged neuronal cells in the hippocampus more than levodopa. These findings suggest that CBD-LNP holds promise as an intervention addressing both metabolic and neurodegenerative aspects of DP, offering a potential therapeutic strategy.

## 1. Introduction

Parkinson’s disease (PD), one of the most common neurodegenerative diseases, affects over 1% of the population over the age of 60, which is estimated to be 10 to 20 cases per 100,000 people [1]. PD stands as a complex and increasing global health issue characterized by motor symptoms such as tremors and impaired coordination [2,3]. These symptoms result from the loss of dopaminergic neurons and the accumulation of Lewy bodies in the substantia nigra, as well as α-synuclein and neuronal degeneration [4,5]. The development of PD poses a challenge due to its complex mechanism involving factors like protein aggregation and neuroinflammation [6]. Besides motor symptoms, PD also exhibits non-motor symptoms, including anxiety, depression, and memory loss [7,8].

Diabetes mellitus, a prevalent chronic metabolic syndrome, affects 9% of individuals globally and increases the risk of PD diagnosis compared to non-diabetic subjects [1]. The coexistence of diabetic Parkinson’s disease (DP) has been gradually reported worldwide with severe symptoms, especially the effect of prolonged high blood glucose levels on synaptic proteins and apoptosis markers in the hippocampus—a critical brain region for memory formation [9,10,11,12]. The lack of understanding of the mechanisms of insulin resistance in diabetes and its interaction with PD is an obstacle to developing effective treatments. As the first-line drug, levodopa was used to compensate for central dopamine levels, but it still has some adverse effects [13]. Therefore, to enhance the efficacy and minimize undesirable side effects, alternative therapeutics, such as natural derivative medicine with innovative formulations, are an ongoing pursuit.

In the realm of alternative therapeutics, cannabinoids, particularly cannabidiol (CBD) derived from *Cannabis sativa*, have gained attention for their neuroprotective properties. This interest extends to pharmacological treatments, such as levodopa, to promote central dopamine. Even though CBD has low affinity for CB1R and CB2R, it can modify the endocannabinoid system and enhance their neuroprotective, anti-inflammatory, and immunomodulatory properties [14]. Furthermore, CBDs also promise to alleviate metabolic dysfunction in diabetes models [15,16,17]. However, CBD natural form still faces challenges such as limited bioavailability, poor water solubility, and less absorption; it is therefore necessary to explore novel formulations to overcome these drawbacks.

Recently, some studies suggested that CBD in nano-formulation, specifically as CBD-loaded lipid nanoparticles (CBD-LNP), can enhance absorption and bioavailability compared to its natural form [18,19]. However, clinical and pre-clinical studies remain limited, especially in DP [20]. This study aimed to assess the potential of CBD-LNP in terms of safety and therapeutic outcomes, including metabolic profiles, behavioral responses, and hippocampal morphology in DP animal models. The study provides a well-rounded understanding of the potential of CBD-LNP as an alternative therapeutic intervention. By enhancing CBD’s bioavailability and stability through nanoencapsulation, the hypothesis was that CBD-LNP could improve locomotor and memory functions in animal models of DP. Ultimately, these findings could significantly contribute to the development of effective treatment strategies, thereby enhancing the quality of life and overall health outcomes for individuals dealing with DP.

## 2. Materials and Methods

### 2.1. Preparation and Physical Stability of Cannabidiol Lipid Nanoparticle (CBD-LNP)

CBD-LNP was prepared using a solvent injection method. Briefly described, lipid nanoparticles were formulated by mixing the solvent and aqueous phases using the microfluidization technique. Isolated CBD powder (Amara Asia Co., Ltd., Bangkok, Thailand) was solubilized in ethanol with lipid components including phosphatidylcholine and cholesterol (Lipoid GmbH, Ludwigshafen, Germany). Then, the solvent phase was mixed with the aqueous phase (deionized water) using a high-speed homogenizer (IKA, Staufen, Germany), and particle size was reduced by microfluidizer (M-110P Microfluidizer, Microfluidics Inc., Westwood, MA, USA). Lipid nanoparticles were formed under mechanical force. Ethanol in the mixture was then removed by rotary evaporation under reduced pressure until 3 mg/mL CBD was obtained.

The encapsulation efficiency (*EE*) of CBD-loaded LNP was determined by using an Amicon membrane filter, followed by centrifugation. Unencapsulated CBD in the aqueous phase (supernatant) was filtered, and the concentration of encapsulated CBD was evaluated and calculated based on HPLC-UV analysis. The encapsulation efficiency was calculated using the following equation:$$\%EE=\frac{C_{i}-C_{f}}{C_{i}}\times 100$$
where:

*C_i_* represents the initial concentration of CBD added to nanoparticles;

*C_f_* represents the concentration of unencapsulated CBD.

The stability of CBD-LNP was confirmed by the average diameter, zeta potential, and polydispersity index (PDI) measured using a Malvern Instruments Zetasizer Nano ZX (Malvern Panalytical Ltd., Malvern, UK), employing the dynamic light scattering (DLS) technique, following storage at 4 °C, 25 °C, and 45 °C for 30 days.

### 2.2. Assessment of Cytotoxicity

Human neuron cells (SH-SY5Y) were maintained in EMEM (Thermo Fisher Scientific, Waltham, MA, USA) supplemented with 10% fetal bovine serum (FBS) (Thermo Scientific), 100 units/mL penicillin, and 100 µg/mL streptomycin (Pen&Strep). Cells were cultured at 37 °C in a 5% CO_2_ incubator. The cytotoxicity of neuron cells was evaluated with the MTT (3-(4,5-dimethylthiazol-2-yl)-2,5-diphenyltetrazolium bromide) tetrazolium reduction assay, which measured mitochondrial succinate dehydrogenase. Cells were seeded at a density of 1 × 10^4^ cells/well in 96-well plates and allowed to grow until 70–80% confluent, followed by treatment with CBD-LNP, blank-LNP, and isolated CBD. Cytotoxicity was examined 24 h post incubation by adding 100 µL MTT solution (1 mg/mL in PBS) to each well followed by incubation for 4 h at 37 °C. MTT solution was removed, and the formazan crystals were dissolved by adding 100 µL DMSO to each well. Finally, the absorbance was measured at 570 nm using SpectraMax M2 microplate readers (Molecular Devices LLC, San Jose, CA, USA).

### 2.3. Animals

Forty adult male Wistar rats (8 weeks old, weighing 180–200 g) were obtained from Nomura Siam International Co., Ltd. (Bangkok, Thailand). The animals were housed in groups of 2–3 rats per cage (*n* = 8/group) in a controlled room maintained at 25 ± 2 °C and 54 ± 5% humidity, with a 12 h light/dark cycle, and were fed a standard chow diet for at least 1 week prior to the start of the experiments (Charoen Pokphand Foods Public Co., Ltd., Bangkok, Thailand). The experiments were carefully planned and conducted to minimize the number of animals used. Body weight, food intake, and water intake were measured daily. For diabetic induction, rats were provided with a high-fat diet (HFD32) ad libitum for 1 week, purchased from CLEA Japan, Inc. (Tokyo, Japan), and were subsequently administered lower doses of by streptozotocin (STZ) injection. After that, rats were maintained on a high-fat diet for 4 weeks. All rats were randomly divided into five equal groups as follows:

(i) Control group + vehicle: Rats were injected with a single intraperitoneal injection of 0.5 mL sodium citrate solution 1 week before the start of the 4-week normal saline injection. Subsequently, rats were orally administered vehicle (3 mL/kg mixed lecithin, phosphatidylcholine, and cholesterol in ethanol).

(ii) DP + vehicle: Diabetic induction was conducted by a single intraperitoneal injection of 35 mg/kg STZ 1 week before the start of 1-methyl-4-phenyl-1,2,3,6-tetrahydropyridine (MPTP)-induced PD (30 mg/kg dissolved in normal saline). Subsequently, rats were orally administered vehicle (3 mL/kg).

(iii) DP + CBD: DP rats were orally administered natural form CBD dissolved in vehicle (20 mg/kg).

(iv) DP + CBD-LNP: DP rats were orally administered CBD-LNP (20 mg/kg).

(v) DP + levodopa (L-dopa; positive control PD agent): DP rats were orally administered levodopa dissolved in normal saline (10 mg/kg).

### 2.4. Induction of Diabetes and Parkinson’s Diseases (DP)

The induction of DP was performed by modifying the methods from Elbassuoni and Ahmed, 2019 [20]. To induce type 2 diabetes mellitus (TDM2), rats were subjected to a high-fat diet for one week and then injected with 35 mg/kg body weight of STZ (Sigma-Aldrich, St. Louis, MO, USA), dissolved in freshly sodium citrate solution (pH 4.5). The STZ injection was administered intraperitoneally as a single dose. Following the STZ injection, the rats were fed high-fat diets for four weeks [21,22]. Blood samples were collected weekly by anesthetizing the rats and extracting blood from the lateral tail veins. The blood glucose levels were analyzed to determine the hyperglycemic condition, and rats with fasting blood glucose levels above 180 mg/dL were considered to have diabetes [20]. For the induction of PD, the diabetic rats were injected with MPTP (Sigma-Aldrich, St. Louis, MO, USA) for 4 weeks. The dosage of MPTP utilized in this study was derived from the research conducted by Li et al. (2023), wherein it was administered intraperitoneally at a dosage of 30 mg/kg over a period of four weeks [23].

### 2.5. Administration of CBD-LNP

According to the preliminary data presented in Appendix A, the administration of 20 mg/kg CBD-LNP had no significant effect on the locomotor activities of normal rats in the open field test (OFT), while concurrently reducing anxiety levels as observed in the elevated-plus maze. This observation is consistent with previous studies conducted by Costa et al., 2007 and Szulc et al., 2013, which demonstrated the potential of CBD in alleviating inflammatory and neuropathic pain, as well as mitigating alcohol tolerance in rats [24,25]. Therefore, considering the positive outcomes reported in these studies, it was decided to orally administer 20 mg/kg CBD-LNP to DP rats.

### 2.6. Administration of Antiparkinsonian Agent

For the administration of levodopa, Levopar tablets containing levodopa and benserazide hydrochloride were obtained from Baxter International Inc., Deerfield, IL, USA. Levodopa is a first-line dopamine replacement agent for the treatment of PD. In the levodopa-treated group, rats were orally administered 10 mg/kg L-dopa between 15:00 and 16:00 once daily for 4 weeks, 7 days a week. Levopar tablets were freshly prepared daily before administration [26,27].

### 2.7. Assessment of Muscular Rigidity-like Symptom

The catalepsy bar test assesses the inability to correct a constrained posture due to muscular rigidity. Rats were gently placed with their forelimbs on a 10 cm high stainless steel bar and hind limbs on the floor. The time taken for paw removal from the bar was measured, with a maximum descent latency set at 180 s. A longer time in the bar test indicates increased muscular rigidity and stiffness in rats [28].

### 2.8. Assessment of Memory-like Behaviors

#### 2.8.1. Y-Maze Test

Memory-like behaviors were assessed using the Y-maze test, a method designed to evaluate spatial working memory [29]. The Y-maze apparatus consisted of three arms diverging at a 120° angle from each other, with a central triangular area. To minimize the influence of stress on behavior, rats were acclimatized to the testing room for at least 30 min prior to the test. During the test, each rat was placed in the Y-maze and allowed eight minutes for free exploration, with its behavioral responses being recorded by an infrared video camera. Rats typically exhibit a tendency to explore recently visited arms and alternate between the three arms. Efficient alternation between arms indicates the utilization of working memory, as rats have to maintain a record of the most recently visited arms [30]. The total number of arm entries, the number of triads (consecutive visits to different arms), and the percentage of alternation were measured in the Y-maze test [31]. Subsequently, after completion of the test, the rats were returned to their home cages. To maintain cleanliness and minimize potential confounding factors between trials, the Y-maze arena was thoroughly cleaned with 20% ethanol. A spontaneous alternation was considered to occur when a rat entered a different arm of the maze in each of three consecutive arm entries. A low percentage of spontaneous alternation is often considered as an indication of impaired spatial working memory [31]. The percentage of alternation was then calculated using the following equation:$$\frac{\mathrm{spontaneous}\;\mathrm{alterations}}{\mathrm{total}\;\mathrm{number}\;\mathrm{of}\;\mathrm{arm}\;\mathrm{entries}-2}\times 100.$$

#### 2.8.2. Novel Object Recognition (NOR) Test

The NOR test was conducted in a black rectangular plastic box measuring 63 cm in length, 63 cm in width, and 45 cm in height, under a room light intensity of 360 lux. The experimental procedure followed the methods described by Lapmanee et al. in 2023. A video camera was positioned above the box to record the behavioral profiles of the rats. The objects used for discrimination were made of glass or ceramic. On the day prior to the NOR test, each rat underwent habituation in the empty box for two sessions, with each session lasting 10 min. During the test, the rat was gently placed in the box and exposed to a 3 min acquisition session with two identical objects (ceramic pepper bottles measuring 3 cm in length, 3 cm in width, and 7 cm in height) positioned approximately 10 cm apart at the center of the box. Following the acquisition session, the rat was returned to its home cage for a 1 h inter-trial interval, during which time the box and objects were cleaned with 20% ethanol. Next, one of the objects in the box was replaced with a novel object (a glass paperweight measuring 5 cm in length, 5 cm in width, and 12 cm in height). The same rat was then placed back in the box and allowed to explore the new object for 3 min. A decrease in the discrimination ratio indicated cognitive and memory impairment [32,33]. The discrimination ratio was calculated using the following equation:$$\frac{\mathrm{Time}\;\mathrm{exploring}\;\mathrm{novel}\;\mathrm{object}\;\left(s\right)-\mathrm{Time}\;\mathrm{exploring}\;\mathrm{familiar}\;\mathrm{object}\;\left(s\right)}{\mathrm{Total}\;\mathrm{exploration}\;\mathrm{time}\;\left(s\right)}.$$

### 2.9. Assessment of Locomotor and Exploratory Activities

The locomotor and exploratory activities of the rats were assessed using the OFT. The apparatus used for the test was constructed from black acrylic plastic and had dimensions of 76 cm in length, 57 cm in width, and 35 cm in height. The floor of the apparatus consisted of a 48-square grid (6 × 8 squares) with each square measuring 9.5 cm per side. The arena was divided into two zones: the inner zone and the outer zone, with the outer zone comprising 24 peripheral squares. To initiate the test, the animal was gently placed in one of the four corner squares of the apparatus and given 5 min to freely explore the environment. The behavioral responses of the rats were recorded using an infrared video camera. An increase in the number of rearing-stretch behaviors, which refers to the rat assuming an upright position, indicated increased exploration. The total number of lines crossed in the grid represented changes in locomotor activity [32,33].

### 2.10. Assessment of Metabolic Parameters

Biochemical analyses of blood were conducted following the completion of the behavioral tests. After anesthesia, the rats were sacrificed and blood samples were immediately collected. The centrifugation at 3000 rpm for serum and 6000 rpm for plasma at 4 °C for 10 min was performed, and then, all blood tubes were kept at −80 °C until analysis. The metabolic parameters consist of body weights, abdominal fat weights, plasma glucose, and lipid profiles (i.e., triglyceride and cholesterol levels). Weekly blood glucose levels were measured using an electronic blood glucose meter in units of mg/dL (Roche Diabetes Care India Pvt Ltd., Mumbai, India), while plasma glucose, triglyceride, and cholesterol levels were analyzed using a Fujifilm biochemical analyzer (Dri-Chem NX500, Tokyo, Japan).

### 2.11. Assessment of Insulin and Insulin Resistance

Insulin levels in the serum samples were determined using the ELISA test kit (catalog number: EZRMI-13K, Merck Millipore, Darmstadt, Germany), following the manufacturer’s instructions. Baseline insulin resistance was measured at week −2, after inducing diabetes at week 0, and after administering the treatments at week 4. Before each time point, the rats were fasted for a minimum of 5 h before blood collection from the tail vein. Insulin resistance and progressive pancreatic β-cell dysfunction have been identified as the two fundamental features in the pathogenesis of TDM2. As a widely validated clinical and epidemiological tool for estimating insulin resistance and β-cell function, the homeostasis model assessment (HOMA) is derived from a mathematical assessment of the balance between hepatic glucose output and insulin secretion from fasting levels of glucose and insulin. The HOMA of insulin resistance (HOMA-IR) index was computed by the following equation:$$\frac{\mathrm{Fasting}\;\mathrm{plasma}\;\mathrm{insulin}\;(\mu U/mL)\times \mathrm{Fasting}\;\mathrm{plasma}\;\mathrm{glucose}\;(mmol/L)}{22.5},$$
while the HOMA of β-cell function (HOMA-β) index was computed by the following equation:$$\frac{20\times \mathrm{Fasting}\;\mathrm{plasma}\;\mathrm{insulin}\;(\mu U/mL)}{\mathrm{Fasting}\;\mathrm{plasma}\;\mathrm{glucose}\;(mmol/L)-3.5}.$$

Conversion factors were insulin (1 μU/mL = 0.0417 ng/mL = 7.175 pmol/L) and blood glucose (1 mmol/L = 18 mg/dL). A higher HOMA-IR and lower HOMA-β index indicate high impaired glucose tolerance [34,35].

### 2.12. Assessment of Striatal Dopamine and Serum Inflammatory Cytokine Levels

The brains were removed and then washed with ice-cold saline. The frontal region of the whole brain, including the striatum area, was freshly dissected. The striata from each brain hemisphere were isolated, weighed, and homogenized in phosphate-buffered saline (pH = 7.4). The resulting homogenate was centrifuged, and the supernatants were collected and stored at −80 °C. Dopamine levels in the striata were determined by using a rat dopamine ELISA kit (catalog number: MBS262606, MyBioSource, San Diego, CA, USA), while cytokine-induced inflammation marker, i.e., tumor necrosis factor (TNF)-α in serum were determined by rat ELISA commercial kits analysis (catalog number: RAB0480, Merck Millipore, Darmstadt, Germany) in accordance with the manufacturer’s instructions.

### 2.13. Assessment of Hippocampal Histomorphological Changes

The brains, including the half middle and back regions of the brain, encompassing the dorsal and ventral hippocampus, were collected and post-fixed in 10% neutral buffered formalin before being processed for histology. Paraffin embedding was conducted, followed by sectioning at 5 μm thickness using a rotatory microtome (Leica, Nussloch, Germany). The sections were then cleared in xylene, hydrated in decreasing alcohols, stained with Hematoxylin and Eosin (H&E) [36,37], and finally mounted with Permount™ mounting media (Fisher Scientific, Geel, Belgium). Subsequently, four specific regions within the hippocampus, including cornu ammonis (CA) 1 to CA 4 and dentate gyrus (DG), were examined to assess neuronal damage. Damaged neurons were identified based on the presence of pyknotic cells and cells exhibiting condensed chromatin. Pyknotic cells were characterized by a densely stained nuclear content that was evenly dispersed throughout. Condensed chromatin was observed when intensely stained nuclei displayed localized regions of aggregated chromatin [38,39]. Neuronal counts were obtained from the hippocampus using a light microscope Nikon DXM 1200 digital camera (Tokyo, Japan) at 400× magnification. For each animal, average neuronal counts were obtained by counting three serial coronal sections. Pyknotic index was calculated as follows:$$\mathrm{Pyknotic}\;\mathrm{index}\;\left(\%\right)=\frac{\mathrm{pyknotic}\;\mathrm{neurons}}{\mathrm{total}\;\mathrm{neurons}}\times 100$$

### 2.14. Statistical Analysis

Statistical analysis was performed using means ± SEM. For comparisons between two sets of data, an unpaired Student’s t-test was used. Multiple comparisons were conducted using one-way analysis of variance (ANOVA) followed by the Dunnett post-hoc test (GraphPad software version 9).

## 3. Results

### 3.1. Successful Preparation, Stability, and Cytotoxicity of CBD-LNP

This study utilized the solvent injection method via microfluidization techniques to formulate nanoparticles. The CBD-LNPs were synthesized from small molecules of phospholipid, cholesterol, and isolated CBD and then investigated for the physicochemical properties (size, charge, and PDI), stability, encapsulation efficiency, and cytotoxicity to neuron cells. The results showed that the synthesized CBD-LNP was obtained with a particle size ≤ 170 nm and consistent size distribution with PDI ≤ 0.2. The zeta potential was −16.57 ± 0.04 as shown in Table 1. The stability test suggests that synthesized CBD-LNP maintained its physicochemical characters over 30 days. CBD was encapsulated in LNP as high as 98.78 ± 0.90% due to the high hydrophobicity of CBD, which tends to encapsulate in the lipid phase.

Further, the cytotoxicity of CBD-LNP, CBD, and vehicle-LNP was evaluated using SH-SY5Y culture cells by incubating them for 24 h. As shown in Figure 1, the results clearly indicated that the toxicity of CBD-LNP was considerably lower than the isolated CBD natural form. It suggests that LNP formulation alleviates the toxicity of CBD from direct contact to neuron cell lines. Moreover, the results obviously demonstrated that CBD-LNP has enhanced neuron cell proliferation at low concentrations due to high cellular uptake, and less toxicity compared with blank-LNP and CBD. These are the advantages of CBD-LNP as a potential treatment to provide benefits in animal studies.

### 3.2. Successful and Validation of the DP Model

The results from behavioral profiles indicated that alterations in motor performance, locomotor activities, and memory behaviors were indicative of the consequences of DP induction in male rats, as shown in the Figure 2A. Figure 2B–G shows that vehicle-treated DP rats exhibited symptoms resembling muscular rigidity, hypolocomotion, lower exploration activity, and memory impairments. This was demonstrated by a significantly increased descent latency on the bar test, decreased total line crosses and rearing numbers in the OFT, as well as a lower percentage of alteration in the Y-maze test, and a lower percentage of total exploration and discrimination index in the NOR test, respectively.

Furthermore, physical and biochemical changes were employed to validate the induction of DP in male rats. There were no significant differences in the starting body weight among the control and test groups. Rats exhibited a slight increase in body weight following 1 week of pretreatment with a high-fat diet and a marked increase in body weight when they received a single low dose of STZ injection. After induction of high-fat and STZ-induced T2DM, rats displayed higher levels of fasting blood glucose and lipid profiles, as well as lower insulin levels compared to the control group. These findings resulted in the impairment of pancreatic function, as demonstrated by significantly increased HOMA-IR and decreased HOMA-β indexes in diabetic rats (Appendix A). Subsequently, the rats were continuously fed a high-fat diet and received daily MPTP injections for 4 weeks to induce both T2DM and PD. Compared to the control group, vehicle-treated DP rats exhibited increased metabolic and partly pancreatic abnormalities, including elevated levels of triglycerides, total cholesterol, fasting blood glucose, decreased insulin, and a lower HOMA-β index. Additionally, they displayed lower striatal dopamine levels and increased systemic inflammation, as indicated by high TNF-α levels. These results indicate that the DP protocols were successful in male rats, as presented in Figure 3.

At the histological level, the control group’s CA1, CA2, CA3, and CA4 regions of the hippocampus displayed glial cells with varying nuclear staining in the pyramidal cell layer (PCL) (Figure 4A–D), while the granule cell layer (GCL) in the DG contained small, rounded granule cell bodies (Figure 4E). Upon closer inspection, vehicle-treated DP rats exhibited disarranged, loosely packed, dark, shrunken pyramidal cell bodies with pyknotic nuclei and pericellular haloes in CA1, CA2, CA3, and CA4 (Figure 4F–I). Additionally, the DG regions showed dark, shrunken granule cell bodies with pyknotic nuclei (Figure 4J). Neuronal cell damage was evident in vehicle-treated DP rats, with an elevated percentage of pyknotic cells and a reduction in the thickness of the PCL and GCL compared to the control group (Table 2). This histological examination revealed disrupted pyramidal and granule cell bodies in vehicle-treated DP rats, indicating successful induction and associated neurological, metabolic, and behavioral alterations.

### 3.3. The Potential Therapeutic Effects of CBD-LNP

Regarding the therapeutic effects of the treatment on muscular rigidity, locomotion, and exploration activity, CBD-LNP was found to reduce muscle stiffness similarly to levodopa, rather than natural CBD, in the bar test (Figure 2B). However, CBD-LNP did not affect locomotor activity in the OFT compared to the vehicle-treated group. Similarly, CBD did not alter locomotor deficits but impaired exploration activity by decreasing the rearing number in the OFT in DP rats compared with controls. As expected, levodopa significantly restored locomotor function in DP rats in the OFT (Figure 2C). Moreover, all treatments attenuated spatial memory impairment in DP rats (Figure 2E–G). CBD-LNP exhibits comparable effects to levodopa, as demonstrated by the increased percentage of alteration in the Y-maze test and exploration activity, as well as the discrimination index in the NOR test. Therefore, CBD-LNP shows promising effects similar to levodopa in mitigating DP-induced motor and memory abnormalities in male rats.

Although none of the treatments strongly attenuated biochemical abnormalities in DP rats compared to control rats, CBD-LNP treatment exhibited effects similar to CBD when compared to controls. However, CBD-LNP treatment provided additional benefits, including improved insulin levels and reduced inflammation without any changes in body weight and fat accommodation compared to vehicle-treated groups. Moreover, CBD-LNP resulted in enhancements in various parameters such as body weight, abdominal fat, blood glucose levels, insulin, pancreatic function, dopamine levels, and inflammation when compared with natural CBD alone.

On the other hand, CBD appears to be less effective in mitigating the detrimental consequences of DP induction. This is demonstrated by decreased final body weight and abdominal fat, along with persistently high levels of lipid profiles, blood glucose, TNF-alpha, and lower HOMA-β and dopamine levels. However, the natural form of CBD could improve metabolic status and dopamine levels compared to the vehicle-treated group. As expected, levodopa did not dominate for modulating effects on metabolic status, including lipid levels, blood glucose, HOMA-IR, and HOMA-β indexes in DP rats compared to control rats. However, levodopa treatment was able to improve striatal dopamine levels and further improve triglyceride and cholesterol levels compared to vehicle-treated DP rats (Figure 3).

Moreover, CBD-LNP demonstrated the potential to repair damaged neuronal cells in DP rats across all subregions of the hippocampus. Treatments with CBD also showed positive results but had a less potent effect on increasing the thickness of the PCL and GCL than CBD-LNP. While levodopa reduced the pyknotic index and increased the thickness of the PCL in CA1, CA3, CA4, and GCL of the DG regions compared to the vehicle-treated DP group; however, the effect was less pronounced in CA2 regions (Table 2). The findings suggest that CBD-LNP demonstrated the potential to repair damaged neuronal cells in the hippocampus more effectively than CBD alone, indicating its promising neuroprotective properties in DP.

## 4. Discussion

The present study, employing the solvent injection method through microfluidization techniques, strives for the successful synthesis of CBD-LNP. This endeavor is supported by compelling scientific findings [40,41,42] suggesting the potential treatment of DP comorbidity. The formulation underwent comprehensive characterization to assess its physicochemical properties, stability, encapsulation efficiency, and cytotoxicity to neuron cells. The nanoparticle size, a crucial determinant for effective drug delivery, was demonstrated to be ≤170 nm with a consistent size distribution (PDI ≤ 0.2). Zeta potential is an important factor in the physical stability of nanoparticles. The higher zeta potential shows better stability of the dispersion [43]. The negative zeta potential of −16.57 ± 0.04 indicates the stability and colloidal nature of the CBD-LNP. These findings align with the established principles of nanoparticle formulation for drug delivery, ensuring optimal size and stability for therapeutic applications [42,44]. The encapsulation efficiency of CBD within the lipid nanoparticles was high at 98.78 ± 0.90%, suggesting a successful incorporation facilitated by the hydrophobic character of CBD [45,46]. This result aligns with the understanding that the lipid phase is conducive to encapsulating hydrophobic compounds, a key aspect of efficient drug delivery systems [47]. The stability test, spanning a 30-day period in three different temperatures, confirms the potential of CBD-LNP for practical applications. The ability to maintain physicochemical characteristics over an extended duration is a crucial aspect in ensuring the feasibility of translating such formulations into therapeutic interventions [48].

Cytotoxicity evaluation in SH-SY5Y neuron cells revealed a significantly lower toxicity profile for CBD-LNP compared to isolated CBD natural form. This suggests that the CBD-LNP formulation relieved the toxicity of CBD on direct contact with neuron cells, indicating a potential safety advantage [45,49]. Moreover, the observed enhanced neuron cell survival at low concentrations, coupled with lower toxicity compared to CBD alone, adds another layer of promise to the potential therapeutic application of CBD-LNP. These findings align with the broader understanding that nanoparticle formulations can enhance drug delivery efficiency and reduce toxicity [49,50]. The findings presented in this study not only demonstrate the successful preparation of CBD-LNP but also highlight its favorable physicochemical properties, stability, and reduced cytotoxicity, positioning it as a promising avenue for further exploration in the treatment of neurological disorders.

Validation of a rat model representing the coexistence of DP indicates the complex mechanism linking metabolic and neurodegenerative disorders [9,20,21,51]. It is noteworthy that diabetes mellitus is independently linked to heightened cognitive impairment in PD [7]. The study employed an integrated approach, including physical and biochemical methods, to evaluate the effects of DP induction on the rat model. A notable observation was the successful establishment of baseline conditions, illustrated by no significant differences in starting body weight between the control and test groups. This foundational information enhances the reliability of later assessments. After one week of pretreatment with a high-fat diet and low-dose STZ injection, the observed increase in body weight, as expected, indicated the successful induction of pre-diabetes in the rat model [21,22,52,53]. The biochemical changes post-induction, including elevated fasting blood glucose levels, reduced insulin levels, and decreased striatal dopamine levels contributing to the development of α-synuclein aggregation, collectively confirm the successful establishment of DP rats [4,20,21,54,55]. The alterations in HOMA-IR and HOMA-β indexes confirmed the development of insulin resistance and impaired β-cell function, providing a comprehensive characterization of the model’s metabolic and neurodegenerative features [34,35,56].

The subsequent treatments involving continuous high-fat diet feeding and daily MPTP injections further emphasized the complexity of the coexisting conditions [20,23,57]. The vehicle-treated DP rats displayed systemic inflammation and metabolic abnormalities, providing a clinically relevant context for assessing therapeutic interventions. Although apoptosis, necrosis, and late apoptosis are interesting aspects of validating the DP model, the established reduction in striatal dopamine levels in the DP group is noteworthy. Additionally, significant decreases have been observed in the activities of striatal antioxidant enzymes, while there have been significant increases in BAX, BCL2-associated apoptosis, and lipid peroxidation, as well as elevated levels of proinflammatory cytokines TNF-α and IL1β. These findings collectively suggest the involvement of oxidative stress, apoptosis, and inflammation in the pathogenesis of DP [20,23,58,59].

Interestingly, CBD treatment demonstrated positive effects, particularly in body weight regulation and adipose tissue accumulation, supporting the potential anti-obesity properties of CBDs [60]. Our study found that the improvements of some parameters in DP rats were observed in CBD, levodopa, and CBD-LNP treatments, including fasting blood glucose levels, lipid, insulin profiles, HOMA-β index, and dopamine levels, which implied the potential of these treatments in restoring both metabolic and neurodegenerative aspects of the coexisting conditions [15,16,17,20,61,62]. The distinct efficacy of CBD-LNP in reducing TNF-α levels and enhancing various metabolic parameters surpasses these effects observed with the CBD natural form, suggesting the increased therapeutic potential of the lipid nanoparticle formulation. This aligns with the emerging field of nanomedicine, where nanoparticles enhance drug delivery efficiency and may contribute to improved treatment outcomes [19,45,63,64].

Additionally, the alteration in motor symptoms and memory-like behaviors in the DP rats indicated the consequence of the induction in DP modeling and presented the efficacy of therapeutic interventions [65]. The assessment of motor symptoms and muscular rigidity revealed significant differences among the treatment groups. The vehicle-treated DP rats exhibited increased descent latency on the bar, indicating the successful induction of PD in this animal model [20]. Interestingly, the treatment with CBD-LNP resulted in a significant decrease in latency on the bar test, suggesting an ameliorative effect on muscular rigidity and muscle stiffness-like symptoms. Likewise, levodopa treatment demonstrated efficacy in alleviating catalepsy, supporting its role as a positive control [66,67].

According to locomotor and exploratory activities, DP rats displayed reduced activity in the OFT, particularly in the vehicle-treated group [20]. Levodopa treatment as a positive control showed increased locomotor activities compared to the vehicle-treated rats. This further supports the effectiveness of levodopa in addressing motor impairments associated with PD [67]. Moreover, all results also suggest that both CBD-LNP and levodopa have the potential to influence motor symptoms in the DP model. Further memory assessments, including spontaneous alternation in the Y maze and discrimination index in the NOR test, provided insight into cognitive function. As a result, vehicle-treated DP rats exhibited lower spontaneous alternation in the Y maze, increased exploration latency, and decreased discrimination index in the NOR test, which indicate memory impairment-like behaviors [30,31,32,68]. In contrast, treatment with CBD, CBD-LNP, or levodopa showed improvements in memory impairment-like behaviors. Specifically, DP rats treated with CBD-LNP exhibited comparative efficacy to levodopa, suggesting that CBD-LNP could be a promising intervention for mitigating cognitive symptoms associated with DP comorbidity. The results from the behavioral assessments, together with the physical and biochemical information, provide a comprehensive understanding of the success of the induced comorbid conditions and the therapeutic effects of CBD-LNP. The comparative efficacy of CBD-LNP to levodopa in addressing both motor and cognitive symptoms illustrates its potential treatment for this complex DP.

Moreover, the observed histopathological changes in the hippocampal subregion, as revealed by H&E staining, provide valuable insights into the neuropathological alterations associated with DP. The higher percentage of pyknotic cells and the reduction in the thickness of the PCL and GLC in vehicle-treated DP rats emphasize the severity of neuronal damage in this condition [12]. This aligns with existing literature suggesting a close interplay between diabetes and neurodegenerative processes, particularly in PD [12,69,70]. The intriguing finding of CBD-LNP’s potential in mitigating damaged hippocampal neuronal cells suggests a neuroprotective effect. This aligns with the known anti-inflammatory and antioxidant properties of CBD, which have been reported in various neurodegenerative conditions [64,71,72]. The specific targeting of damaged regions by CBD-LNP is noteworthy and warrants further exploration to elucidate the underlying mechanisms responsible for this observed neuroprotection. The contrasting impact of levodopa, a standard treatment for PD, raises questions about its efficacy in DP [73]. The increased percentage of pyknotic cells following levodopa treatment in the hippocampus of DP rats suggests the need for careful consideration and evaluation of traditional PD treatments in the presence of comorbid diabetes [74,75].

Collectively, these findings suggest the presence of neuronal injury in the DP model and highlight the potential of CBD-LNP as a therapeutic intervention. However, it is crucial to address the complexity of these interactions and regard these observations as preliminary for clinical research. While animal models provide valuable insights, they may not fully capture the intricacies and variability of human diseases. Moreover, the study’s focus on short-term assessments over a 30-day period might not adequately capture the long-lasting effects of CBD-LNP treatment. To validate the superior effects of CBD-LNP at levels higher than those of natural CBD, assessing oral bioavailability would provide a more comprehensive understanding of the effects observed in the present study.

The present study strictly adhered to animal ethics procedures and aimed to minimize their suffering while reducing the number of animals used for tests. A group of six animals was deemed minimal for ethical statistical analysis related to the PD model [23], while our study assigned eight rats to each group. In addition, the identification of specific glial cells indicating neuroimmune activation could provide the characterization of the neuroinflammatory process and evaluate the potential attenuation by CBD-LNP. Furthermore, exploring the amount of proliferating neurons using neurogenesis markers should be further investigated.

Therefore, further investigations should delve deeper into the molecular and cellular mechanisms, particularly exploring CBD-LNP’s neuroprotective effects and its implications for long-term neurodegenerative processes in DP.

## 5. Conclusions

In summary, the results of this study have demonstrated that CBD encapsulated in lipid nanoparticles can alleviate DP symptoms in an animal model, which has never been published before. This includes improvements in metabolic abnormalities, memory impairment-like behaviors, locomotor and exploratory activities, as well as the suppression of neuroinflammation and the restoration of the hippocampal histological architecture. Based on these results, this innovative nano-formulation holds promise as a potential treatment for addressing neuronal degeneration associated with DP disease. However, it is important to note that the precise therapeutic mechanism and dosage of CBD-LNP have not yet been disclosed. Hence, future additional pre-clinical studies are still necessary.

## Figures and Tables

**Figure 1 pharmaceutics-16-00514-f001:**
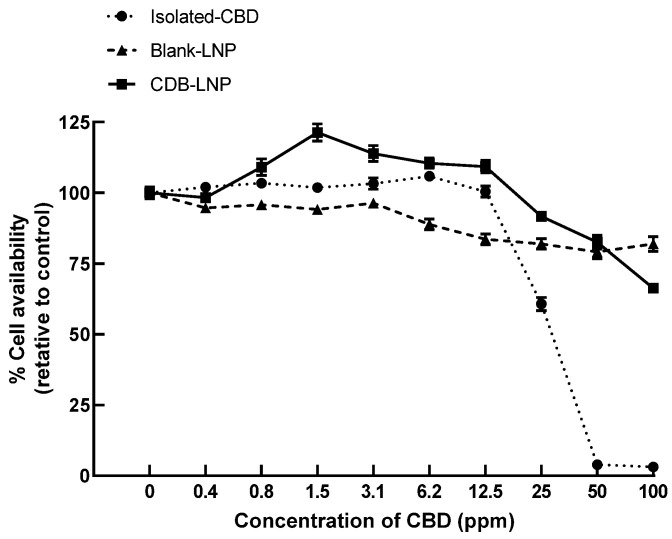
The cytotoxicity of CBD-LNP, blank-LNP, and isolated CBD at various concentrations to neuron cells (SH-SY5Y), incubated over 24 h.

**Figure 2 pharmaceutics-16-00514-f002:**
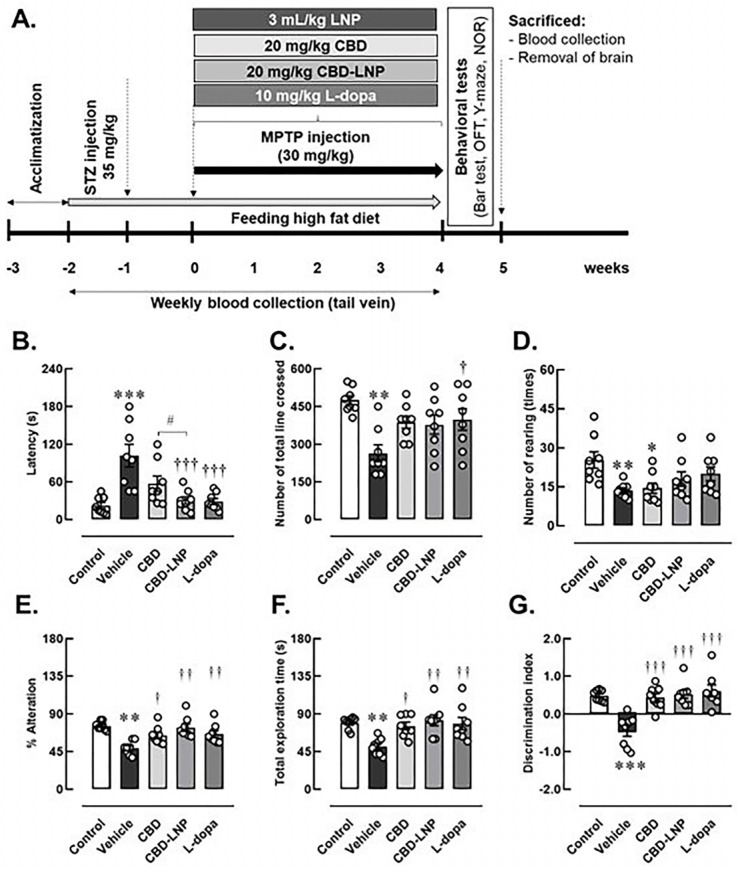
Experimental animal study (**A**) and changes of physical activities and memory profiles after 4 weeks of vehicle, CBD, CBD lipid nanoparticles (CBD-LNP), or levodopa (L-dopa) treatments in diabetic Parkinson’s disease (DP) rats as demonstrated by (**B**) bar test, (**C**) total lines crossed, (**D**) number of rearings in open field test, (**E**) the percentage of alternation in Y-maze test, (**F**) total exploration time, and (**G**) discrimination index of novel objective recognition test. Data are presented as mean ± SEM (n = 8 rats/group). * *p* < 0.05, ** *p* < 0.01, *** *p* < 0.001 compared to vehicle-treated control group. † *p* < 0.05, †† *p* < 0.01, ††† *p* < 0.001 compared to vehicle treated-DP group, and # *p* < 0.05 compared to CDB treated-DP group.

**Figure 3 pharmaceutics-16-00514-f003:**
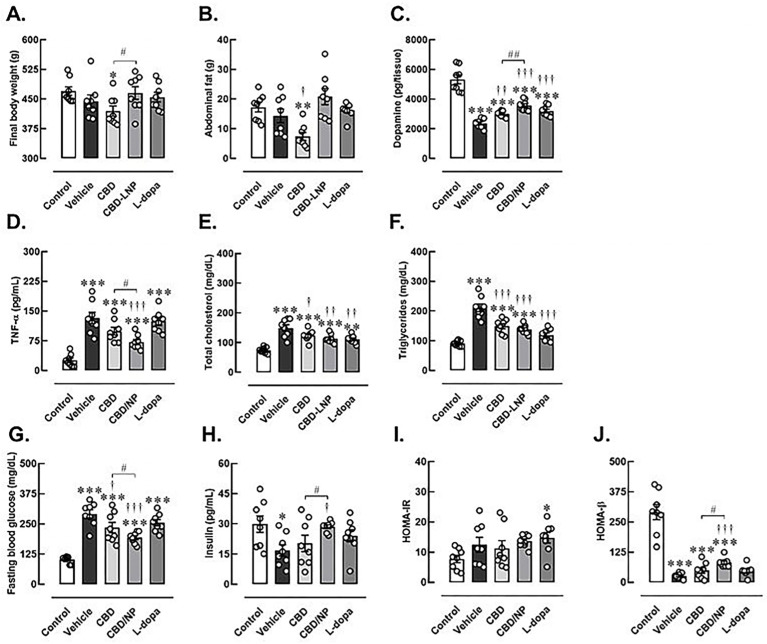
Changes of physical and biochemical profiles after 4 weeks of vehicle, CBD, CBD lipid nanoparticles (CBD-LNP), or levodopa (L-dopa) treatments in diabetic Parkinson’s disease (DP) rats as determined by neuroinflammatory and metabolic biomarkers. (**A**) Final body weight, (**B**) abdominal fat weight, (**C**) striatal dopamine, (**D**) proinflammatory TNF-α, (**E**) total cholesterols, (**F**) triglycerides, (**G**) fasting blood glucose, (**H**) insulin, (**I**) HOMA-IR (Homeostatic Model Assessment for insulin resistance), and (**J**) HOMA-β (Homeostatic Model Assessment estimates steady state beta cell function). Data are presented as mean ± SEM (n = 8 rats/group). * *p* < 0.05, ** *p* < 0.01, *** *p* < 0.001 compared to control, † *p* < 0.05, †† *p* < 0.01, ††† *p* < 0.001 compared to vehicle treated-DP group and # *p* < 0.05, ## *p* < 0.01 compared to CDB treated-DP group.

**Figure 4 pharmaceutics-16-00514-f004:**
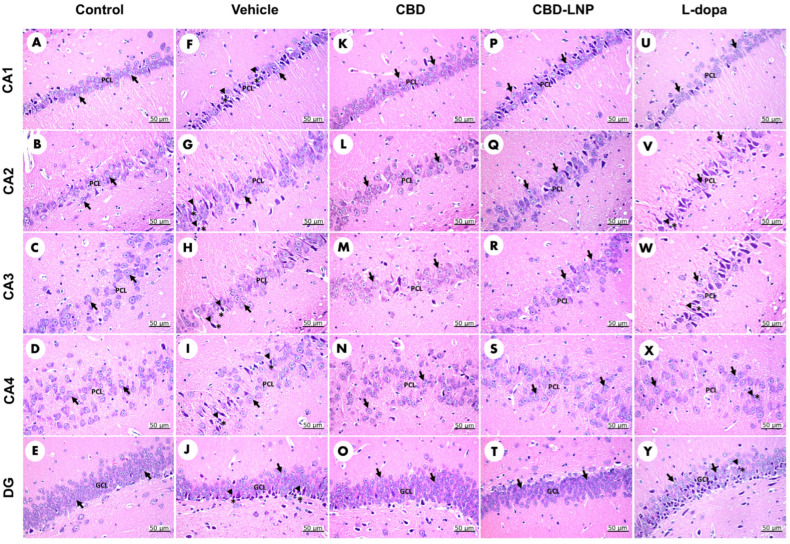
Morphological evaluation of H&E-stained sections in the CA1, CA2, CA3, CA4, and dentate gyrus regions of the hippocampus in diabetic Parkinson’s disease (DP) rats. Control rats (**A**–**E**) display normal neuron cells (arrow). Vehicle-treated DP rats (**F**–**J**) exhibit neuron cells (arrow) alongside highly condensed pyknotic nuclei (head arrow) and pericellular haloes (asterisk). CBD-treated DP rats (**K**–**O**) also show normal neuron cells (arrow). CBD-LNP-treated DP rats (**P**–**T**) reveal numerous normal neuron cells (arrow) with reduced pyknotic nuclei (head arrow) and pericellular haloes (asterisk). Levodopa-treated DP rats (**U**–**Y**) display normal neuron cells (arrow) along with pyknotic nuclei (head arrow) and pericellular haloes (asterisk). Scale bars represent 50 µm (magnification, 40×). PCL, pyramidal cell layer; GCL, granule cell layer.

**Table 1 pharmaceutics-16-00514-t001:** Initial hydrodynamic size, polydispersity index, and zeta potential of blank-LNP and CBD-LNP after being kept at 4, 25, and 45 °C for 30 days.

**Hydrodynamic Size (nm)**	**Day 0**	**Day 30**		
		**Kept at 4 °C**	**Kept at 25 °C**	**Kept at 45 °C**
Blank-LNP	169.93 ± 34.41	163.50 ± 3.78	151.27 ± 2.15	153.80 ± 0.44
CBD-LNP	156.33 ± 1.24	158.17 ± 1.06	167.87 ± 0.81	166.83 ± 5.37
**Polydispersity**	**Day 0**	**Day 30**		
		**Kept at 4 °C**	**Kept at 25 °C**	**Kept at 45 °C**
Blank-LNP	0.41 ± 0.02	0.18 ± 0.02	0.10 ± 0.03	0.07 ± 0.03
CBD-LNP	0.11 ± 0.10	0.12 ± 0.02	0.10 ± 0.03	0.13 ± 0.02
**Zeta Potential (mV)**	**Day 0**	**Day 30**		
		**Kept at 4 °C**	**Kept at 25 °C**	**Kept at 45 °C**
Blank-LNP	−3.53 ± 0.15	−15.28 ± 0.09	−17.89 ± 0.84	−13.04 ± 0.63
CBD-LNP	−16.57 ± 0.04	−17.04 ± 0.90	−14.47 ± 2.08	−19.55 ± 0.70

**Table 2 pharmaceutics-16-00514-t002:** Pyknotic index, pyramidal cell layer in the hippocampus (CA1, CA2, CA3, and CA4), and granular cell layer in the dentate gyrus after 4 weeks of vehicle, CBD, CBD lipid nanoparticles (CBD-LNP), or levodopa (L-dopa) treatments in diabetic Parkinson’s disease (DP) rats.

Parameters	Control Group	DP Group			
	Vehicle	Vehicle	CBD	CBD-LNP	L-Dopa
Pyknotic index of the CA1 (%)	7.70 ± 3.25	22.59 ± 12.91 **	11.50 ± 4.30 †	9.78 ± 5.01 ††	7.86 ± 0.84 ††
Pyknotic index of the CA2 (%)	6.13 ± 1.67	40.32 ± 14.39 ***	8.86 ± 3.94 ††	12.45 ± 5.61 ††	34.39 ± 10.39 **
Pyknotic index of the CA3 (%)	6.78 ± 0.97	38.96 ± 10.41 **	14.00 ± 0.86 †	6.12 ± 2.41 ††	14.21 ± 4.48 †
Pyknotic index of the CA4 (%)	7.84 ± 3.82	55.95 ± 9.98 **	8.22 ± 2.24 ††	14.12 ± 7.04 ††	21.61 ± 15.31 †
Pyknotic index of the DG (%)	6.39 ± 3.76	25.83 ± 8.51 **	10.04 ± 0.84 †	11.05 ± 4.90 †	14.51 ± 5.73 †
Thickness of the pyramidal cell layer in the CA1 (µm)	47.28 ± 1.19	31.99 ± 1.72 ***	47.47 ± 1.95 †††	48.51 ± 1.45 †††	45.8 ± 1.26 †††
Thickness of the pyramidal cell layer in the CA2 (µm)	55.69 ± 1.98	47.67 ± 2.27 ***	51.28 ± 1.52 †	52.89 ± 1.83 ††	48.48 ± 1.44 *
Thickness of the pyramidal cell layer in the CA3 (µm)	57.47 ± 1.15	39.97 ± 1.30 ***	51.19 ± 1.32 *†††	54.45 ± 1.71 †††	51.72 ± 1.36 *†††
Thickness of the pyramidal cell layer in the CA4 (µm)	93.29 ± 3.33	70.16 ± 2.38 ***	82.63 ± 3.56 †	88.23 ± 3.42 †††	89.10 ± 2.17 †††
Thickness of the granular cell layer in the DG (µm)	64.07 ± 1.91	50.14 ± 1.73 ***	58.36 ± 1.08 ††	61.08 ± 1.62 †††	62.18 ± 1.15 †††

Data are presented as mean ± SEM (n = 3–4 rats/group). * *p* < 0.05, ** *p* < 0.01, *** *p* < 0.001 compared to vehicle-treated control group. † *p* < 0.05, †† *p* < 0.01, ††† *p* < 0.001 compared to vehicle treated-DP group.

## Data Availability

The data presented in the study are available on request from the corresponding author.

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
