# Peer review of "Assessing the Safety and Therapeutic Efficacy of Cannabidiol Lipid Nanoparticles in Alleviating Metabolic and Memory Impairments and Hippocampal Histopathological Changes in Diabetic Parkinson’s Rats"

_pharmaceutics, 2024, doi:10.3390/pharmaceutics16040514_

Round 1

Reviewer 1 Report

Comments and Suggestions for Authors

The manuscript by [Sarawut Lapmanee] et al, submitted to Pharmaceutics “Assessing the safety and therapeutic efficacy of cannabidiol Lipid nanoparticles in alleviating metabolism and memory impairment, and hippocampal histopathological changes in diabetic Parkinson's rats”. The study investigates the potential of cannabidiol lipid nanoparticles (CBD-LNP) in animal models of diabetic Parkinson's disease (PD), addressing metabolic parameters and histopathology. The results indicate that CBD-LNP may have beneficial effects in reducing lipid profiles, improving insulin secretion and restoring striatal dopamine levels, with efficacy comparable to levodopa in improving motor and memory deficits. Furthermore, CBD-LNP showed potential in mitigating damage to neuronal cells in the hippocampus, suggesting its promise as a potential therapeutic strategy.

Points to consider

- The introduction could include the epidemiology of Parkinson's disease and diabetes, to indicate the importance of the study.

- The study would need to include the evaluation of the stability of the nanoparticles.

- The assessment of viability by MTT is an exploratory analysis, but it would be interesting to assess apoptosis, necrosis, late apoptosis, this can be done, for example, using the flow cytometry technique.

- It would be better to put the data in table 2 in graphic form so that the reader can better understand the representation of the parameters;

- The scale of figure 3 could be increased and it is possible to use some indicators as arrows within the figure.

- In discussions about the work, the limitations of the work would need to be mentioned.

- It would need to include a figure where it is easier to understand the experimental design, including the groups, evaluation techniques, among other data.

Comments on the Quality of English Language

Need small corrections to the writing in the manuscript

Author Response

- The introduction could include the epidemiology of Parkinson's disease and diabetes, to indicate the importance of the study.

Thank you for your valuable suggestions. We have already improved the introduction part to include the epidemiology of Parkinson's disease and diabetes, as shown on pages 1 and 2.

- The study would need to include the evaluation of the stability of the nanoparticles.

Thank you for your valuable suggestions. We have already added the evaluation of the stability of the nanoparticles in the Materials and Methods section in the Preparation and physical stability of cannabidiol lipid nanoparticle (CBD-LNP) on page 2-3.

- The assessment of viability by MTT is an exploratory analysis, but it would be interesting to assess apoptosis, necrosis, late apoptosis, this can be done, for example, using the flow cytometry technique.

Thank you for your valuable suggestions. We are currently focusing on apoptosis, necrosis, and late apoptosis in our study. However, we were unable to conduct the experiment, so our future studies will delve deeper into this area. We have already included this limitation in our discussion section on page 13.

- It would be better to put the data in table 2 in graphic form so that the reader can better understand the representation of the parameters;

Thank you for your valuable suggestions. We have already adjusted Table 2 to Figure 3 in order to improve readability, as shown on page 11.

- The scale of figure 3 could be increased and it is possible to use some indicators as arrows within the figure.

Thank you for your valuable suggestions. We have already added arrow indicators and adjusted the figure as shown on page 11.

- In discussions about the work, the limitations of the work would need to be mentioned.

Thank you for your valuable suggestions. We have already mentioned the limitations of this study and opportunities for improvement in further studies, as shown on pages 14-15.

- It would need to include a figure where it is easier to understand the experimental design, including the groups, evaluation techniques, among other data.

Thank you for your valuable suggestions. We have already rearranged the experimental design in Figure 2, as shown on page 9.

Reviewer 2 Report

Comments and Suggestions for Authors

This manuscript by Lapmanee et al, describes the effect of a newly generated cannabidiol (CBD) nanoformulation (for oral administration) on in vitro cell proliferation/viability, and plasma metabolic profiles, cognitive and motor performance, and brain histopathology in a diabetic parkinson’s disease (DP) rat model. While vehicle treated DP rats exhibited metabolic abnormalities, decreased striatal dopamine levels, and motor and memory deficits, treatment with CBD nanoformulation but not naturally available CBD reversed these changes. Moreover, CBD nanoformulation in addition to showing similar efficacy to levodopa in reducing motor deficits and memory impairment also promoted replacement of damaged neurons in the hippocampus much better than levadopa. The results are exciting, very interesting and have immense translational potential. However, additional information/evidence is needed to make the content and findings more convincing to the reader and enhance the overall impact and translatability of the manuscript. 

 1.     Lines 60-61: Unlike THC, CBD does not exert its biological effects through CB1 and CB2 receptors. CBD is known to be an allosteric negative modulation of CB1 receptors. This is incorrect and the authors should refer to the literature to include the correct information here.

 2.     Although CBD-LNPs produced beneficial effects, there is no evidence provided to convince the reader that these effects were achieved via increasing the bioavailability of CBD in blood through nanoformulation. It is well known that the bioavailability of orally ingested cannabinoids is only around 2%. Hence, plasma CBD-LNP data should be provided to support the superior mode of action of CBD-LNPs and that levels higher than the natural form of CBD was achieved in plasma with nanoformulation. Otherwise, it could be seen as a placebo like effect.

3.     In Figure 3, in addition to the histopathological findings, the authors should use glial cell specific markers to identify the specific glial cell types (astrocytes vs oligodendrocytes vs microglia) and quantify their numbers and use activation markers for these cells to determine if there is neuroimmune activation and whether CBD-LNPs attenuated activation. Neuron specific markers should also be used to stain and quantify neurons in at least 8-10 fields and show that their numbers are reduced in vehicle treated animals compared to those receiving CBD-LNPs or CBD or levodopa.

4.     In Figure 3, A higher magnification image should be provided showing the degenerating neurons, especially, those showing loss of Nissl’s substance, pyknotic nuclei and perinuclear vacuolization etc.

 5.     Another important piece of information missing is evidence showing neuronal cell proliferation to replace MPTP induced neuronal loss in the hippocampus of CBD-LNP treated DP rats. This must be done using bromodeoxy uridine (Brdu) injections given before necropsy so that proliferating neurons will take up this marker during cell division and fluoresce green that can be detected in postmortem brain sections. 

 6.       There are numerous typographical and grammatical errors throughout the manuscript. Numerous sentences in the manuscript are poorly structured and make it hard to understand the authors interpretation of the findings. It is highly recommended that the authors hire an English language expert/writer to fix the errors before resubmission. 

Comments on the Quality of English Language

  The manuscript is riddled with numerous typographical and grammatical errors. Numerous sentences in the manuscript are poorly structured and make it hard to understand the authors interpretation of the findings. It is highly recommended that the authors hire an English language expert/writer to fix the errors before resubmission.

Author Response

We apologize for this mistake and have already corrected and added it to the introduction part as shown on page 2

Regarding the concerns raised in topics 2-5, we appreciate the insightful suggestions provided by the reviewer and will ensure to address these limitations and considerations in the discussion part of our manuscript.

Reviewer 3 Report

Comments and Suggestions for Authors

The work is interesting and raises an important social problem of complications of a lifestyle disease such as diabetes, so I congratulate the authors of the selected topic. The text of the work, including the introduction and methods, should be more extensive to familiarize the reader with the topic. I would suggest adding epidemiological data on diabetes and its complications, including Parkinson's disease. Also, not all abbreviations are expanded inside the text, so pay attention to this when making corrections, e.g.: STZ.

The results are described in detail and the discussion is exhaustive. It should be clearly noted that the group is very small, which is appropriate for bioethical reasons, but still hypothetical in terms of the results obtained.

This should be clearly stated.

The work may be published after small corrections in the text.

Author Response

Thank you for your positive feedback. We appreciate your suggestions for improving the manuscript. We have expanded the text, especially in the introduction and methods sections, to provide more extensive information on epidemiological data related to diabetes and its complications, including Parkinson's disease. Additionally, we will ensure that all abbreviations are properly expanded within the text, including "STZ." Regarding the group size, we acknowledge the bioethical considerations that led to a small group size. We have emphasized this point in our discussion to highlight the potential limitations on pages 14-15.

Round 2

Reviewer 2 Report

Comments and Suggestions for Authors

In the revised manuscript, the authors have addressed some of the minor critiques. However, major concerns such as providing plasma CBD levels and using glial and neuronal markers to quantify changes in the glial and neuronal cell activation have not been addressed. Moreover, I agree that it is not possible at this stage to do Brdu studies to confirm the ability of CBD to induce neuronal proliferation from neural stem cells. However, showing plasma CBD levels in both CBD-LNP, CBD treated and untreated groups is critical to support the authors claims that the beneficial effects described in the paper occurred as a result of increased plasma CBD bioavailability achieved through LNP mediated delivery.  This is critical since the CBD-LNPs were administered orally, a delivery route that results in very low bioavailability. Again, showing histopathological images does not support changes in neuroimmune activation, a hallmark of neurodegenerative diseases. Since it is customary to collect brain tissues in paraffin embedded blocks, immunofluorescence studies using glial cell markers should be performed to show the effect of CBD on glial cell activation in DP rats.

Comments on the Quality of English Language

Grammatical issues still persist in the revised version. The authors should consult an English language expert to improve the readability of their manuscript.